# Adaptive experimental design produces superior and more efficient estimates of predator functional response

Nikos E. Papanikolaou[1,2†], Hayden Moffat[3], Argyro Fantinou[5], Dionysios P. Perdikis[1], Michael Bode[3], Christopher Drovandi[3,4]*

1 Laboratory of Agricultural Zoology and Entomology, Department of Crop Science, Agricultural University of Athens, Athens, Greece, 2 Department of Plant Protection Products, Hellenic Ministry of Rural Development and Food, Athens, Greece, 3 School of Mathematical Sciences, Queensland University of Technology, Brisbane, QLD, Australia, 4 Centre for Data Science, Queensland University of Technology, Brisbane, QLD, Australia, 5 Laboratory of Ecology and Environmental Science, Department of Crop Science, Agricultural University of Athens, Athens, Greece

† Deceased.
* c.drovandi@qut.edu.au

**Data Availability Statement:** Data and computer code are available in the public GitHub repository (https://github.com/cdrovandi/Adaptive-Design-Functional-Response).

## Abstract

Ecological dynamics are strongly influenced by the relationship between prey density and predator feeding behavior—that is, the predatory functional response. A useful understanding of this relationship requires us to distinguish between competing models of the functional response, and to robustly estimate the model parameters. Recent advances in this topic have revealed bias in model comparison, as well as in model parameter estimation in functional response studies, mainly attributed to the quality of data. Here, we propose that an adaptive experimental design framework can mitigate these challenges. We then present the first practical demonstration of the improvements it offers over standard experimental design. Our results reveal that adaptive design can efficiently identify the preferred functional response model among the competing models, and can produce much more precise posterior distributions for the estimated functional response parameters. By increasing the efficiency of experimentation, adaptive experimental design will lead to reduced logistical burden.

## Introduction

The rate at which prey are consumed by predators—the predator's "functional response"—is a key element in accurate predator-prey and ecosystem modelling [1]. Functional responses are classically classified into three main types (Fig 1) by their shape [2]. In type I there is a (piecewise) linear relation between prey density and the number of prey killed: twice the density of prey results in twice as many prey consumed. In type II prey consumption, predators must expend time "handling" each prey they find, and so the number of prey being consumed increases asymptotically with density. A type III functional response is characterized by a sigmoidal response to increasing prey density, and is generally explained by the predators learning more efficient ways to consume. Types II and III are density-dependent, a fact that seems

**Funding:** Christopher Drovandi was supported by
an Australian Research Council Discovery Project
(DP200102101).

**Competing interests:** The authors have declared
that no competing interests exist.

to prevail in nature, while a type I functional response may be an acceptable approximation for certain types of simple predation, such as filter-feeding [3].

Holling's system offers an elegant categorisation of an important ecological process, and was almost immediately incorporated into the ecological canon [4]. Because of their simplicity and reasonable mechanistic explanations, Holling's three functional forms have been employed throughout theoretical and experimental ecology. An understanding of predator behavioral response to increasing prey density provides deeper insights into predator-prey dynamics. For example, type II functional responses are known to destabilize predator-prey systems as a result of high prey consumption at low prey densities, while type III may promote stability by providing prey refuge at low densities or as a result of predator switching prey [5–7]. Functional responses allow predator foraging efficiency to be evaluated (e.g. [8–10]), predict the magnitude of trophic cascades following climate change (e.g. [11–13]), predict the success of invasive species (e.g. [14–16]) and their subsequent eradication [17], or examine intra- and interspecific interactions (e.g. [18–21]). To answer each of these questions in applied and theoretical ecology, understanding the type of predator functional response, and estimating its parameters, can be critical.

Recently, [22] cast doubt on the methods that ecologists have been using to experimentally determine the functional response type. At the heart of their critique is the lack of information content available from data collected using typical experimental design strategies. Traditional

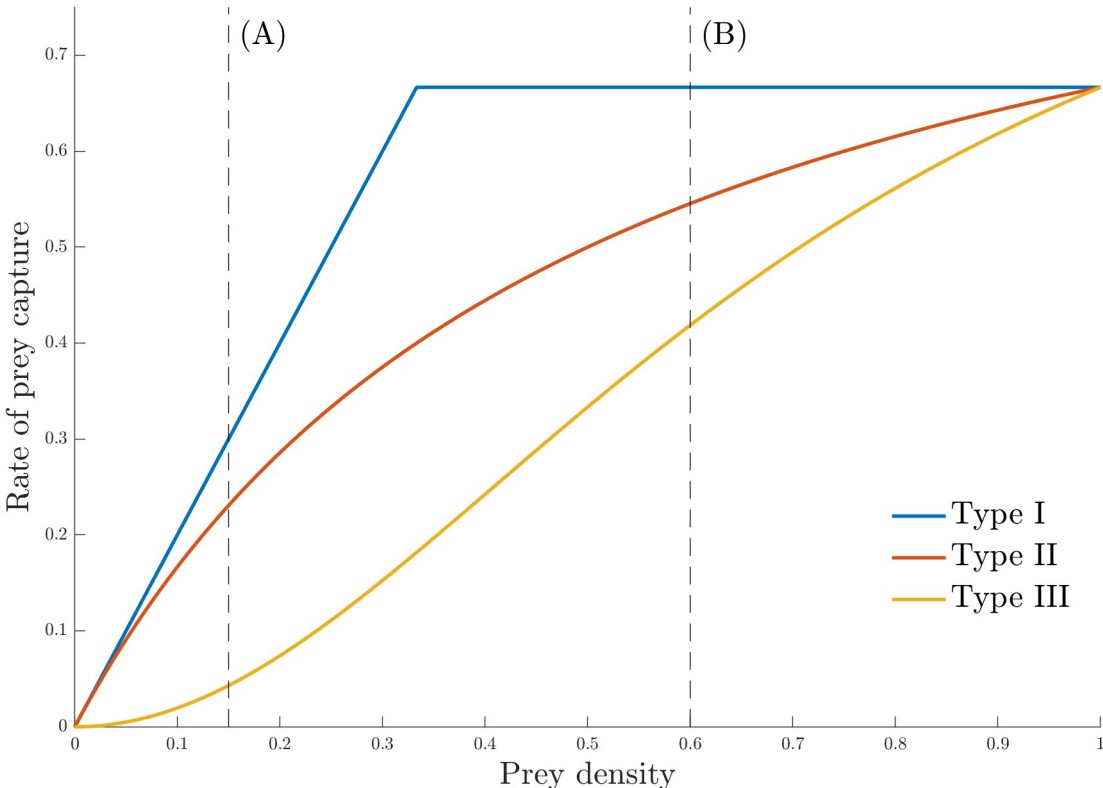

**Fig 1. Illustration of Holling's three predator functional response types.** As the density of the prey increases (x-axis), the rate of prey capture increases (y-axis), at different rates. The vertical dash lines marked (A) and (B) represent prey densities at which it is easy and difficult, respectively, to discriminate between functional response of types I-III. Such prey densities can be easily identified when the parameters governing each functional response is known, which is generally not the case.

experimental design for functional response trials provides a single predator (or multiple predators) with access to fixed numbers of prey for a given period of time (see [23] for a summary report of functional response studies). The consumed number of prey out of the total that the predator has access during that given time period is recorded. This process is then repeated independently of previous experiments, either in parallel or sequentially as resources permit, for a total of $T$ experiments. Each individual experiment is assigned a given initial prey density from a set of initial prey densities to be tested. In this paper we refer to such a data collection process as a non-optimal experimental design. However, there is no assurance that the chosen prey levels and number of independent replicates will be optimal in any way from a statistical point of view. In Fig 1, for example, experiments undertaken at the prey level marked (A) can more easily differentiate the Type III functional response from types I and II. By contrast, experiments undertaken at the prey level marked (B) can more easily differentiate the Type I functional response. As experimental data begins to accumulate, it is possible that one of these two levels of prey density offers more information; a non-optimal experimental design cannot adapt to this fact.

Given the challenges involved in collecting experimental data, and of differentiating between similar functional forms, experimental design offers a useful framework for maximising the statistical benefits offered by each experimental trial [24]. [25] developed an approach to optimally select the prey densities for $T$ experiments according to a statistical criterion that aims at accurate parameter estimation. Since this process is performed prior to any experimentation, we refer to this as an optimal static experimental design. Recently, [26] proposed an optimal adaptive experimental design method, that chooses a prey level for each subsequent experiment based on the data collected from preceding experiments, in order to quickly identify the correct functional response, as well as precisely estimate its parameters using the lowest number of experiments. Using computer simulations, [26] illustrated that far fewer experiments would be required to achieve these goals compared to static experimental design or non-optimal experimental design. In this paper, our goal is to provide concrete evidence that a principled statistical approach is a useful tool for ecologists to identify the correct functional response type and precisely estimate its parameters with much less experimentation effort than currently used methodologies. We put this claim to an empirical test using microcosm ecosystems that is the most common type of experimentation used in these kind of studies. The results support inferences in community ecology, and encourage the application of these methods more broadly across the field.

## Materials and methods

### Functional response data

Microcosm experiments were conducted using *Macrolophus pygmaeus* Rambur (Hemiptera Miridae) preying on eggs of *Ephestia kuehniella* Zeller (Lepidoptera: Pyralidae) as the predator-prey system. Experimental arena consisted of Petri dishes (9 cm diameter, 1.5 cm height) with a mesh-covered hole in the lid (3 cm diameter) to reduce the accumulation of humidity. A tomato leaflet (about 6.50 cm in length and 3.50 cm in width) was placed, abaxial surface up, on a layer of water-moistened cotton wool on the bottom of each Petri dish. On the tomato leaflet eggs were placed at different numbers according to the different experiments, as will be described in the next section. The eggs were distributed randomly on the leaflet. One female or one male of the predator was introduced in each dish. The dishes were kept under controlled conditions of 25˚C, 65 ± 5% RH and 16 h light per day. After 8 hours, the predators were removed from the dishes and the number of eggs consumed was recorded in each dish.

We contrast the performance of our adaptive experimental design approach with a baseline experimental design, which we define as a sensible experimental design chosen by an expert in the predator-prey system under investigation. This is an example of a non-optimal static experimental design. In the baseline experimental design we perform 8 experiments (i.e. one predator in a dish) at 5 different prey levels (2, 5, 10, 20, 40 eggs per dish), which produces 40 experiments for males and 40 experiments for females. These prey levels reflect a reasonable *a priori* experimental design for this application.

### Statistical approach for selecting prey densities

We first provide details on the statistical models commonly used to analyse functional response experimental data. Assume that a single experiment has been conducted with a time interval of $\tau$ and prey density $N_0$ available at the start of an experiment. Denote $N_\tau$ as the prey density at time $\tau$. Holling's type II model (commonly known as the disc equation) describing how $N_\tau$ evolves is given by the following ordinary differential equation:

$$\frac{dN_\tau}{dt} = -\frac{\alpha N_\tau}{1 + aT_h N_\tau},$$

where $\alpha$ and $T_h$ represent the attack rate, i.e. the per capita prey consumption at low prey densities, and the handling time, i.e. the time a predator spends subduing, pursuing and eating a prey item, respectively. A particular form of Holling's type III model that we consider in this paper is captured by the following ordinary differential equation:

$$\frac{dN_\tau}{dt} = -\frac{\alpha N_\tau^2}{1 + aT_h N_\tau^2}.$$

Although there are other models available in the literature (e.g. [27]), we focus on the Holling's type II and III models for their popularity and simplicity. We do not consider Holling's type I model as it is too simplistic for the predator-prey system we consider in this paper, but it can be easily incorporated into the list of candidate models if desired. For the above deterministic models, $p_\tau = (N_0 - N_\tau)/N_0$ represents the proportion of prey consumed by time $\tau$. However, the nature of predator attacks are probabilistic, and we thus link the deterministic model to a probabilistic model. Assume that the number of prey consumed during the time period $\tau$ is $n$, i.e. the recorded observation, which is treated as a random variable. A natural assumption is to model $n$ as being binomially distributed, i.e. $n \sim binomial(N_0, p_\tau)$, where now the interpretation of $p_\tau$ is that it is the probability that a single prey has been consumed by time $\tau$. However, data from typical predator-prey experiments often have more variation than can be explained by the binomial distribution (i.e. overdispersion). To model this overdispersion, a beta-binomial distribution has been considered (e.g. [25, 26, 28]), which includes an additional parameter $\lambda$ to capture the overdispersion. (In the S1 Appendix, we show that a beta-binomial model still performs well if the underlying process is binomial).

To precisely estimate the parameters and discriminate between the models, multiple independent experiments are needed, collected either sequentially or in parallel, with a potentially different prey density used for each individual experiment. For all experiments we assume the same $\tau$ is used. We assume that at most $T$ independent experiments can be conducted, and we denote the prey densities used for these experiments as $N_0^{1:T} = (N_0^1, N_0^2, \ldots, N_0^T)$ where $N_0^t$ is the prey density for observation $t$ and the $T$ experiments are conducted independently. In functional response experiments, the experimenter can often choose the value $N_0^{1:T}$. We denote the corresponding collected data, the number of prey attacked for each experiment, as $n^{1:T} = (n^1, n^2, \ldots, n^T)$.

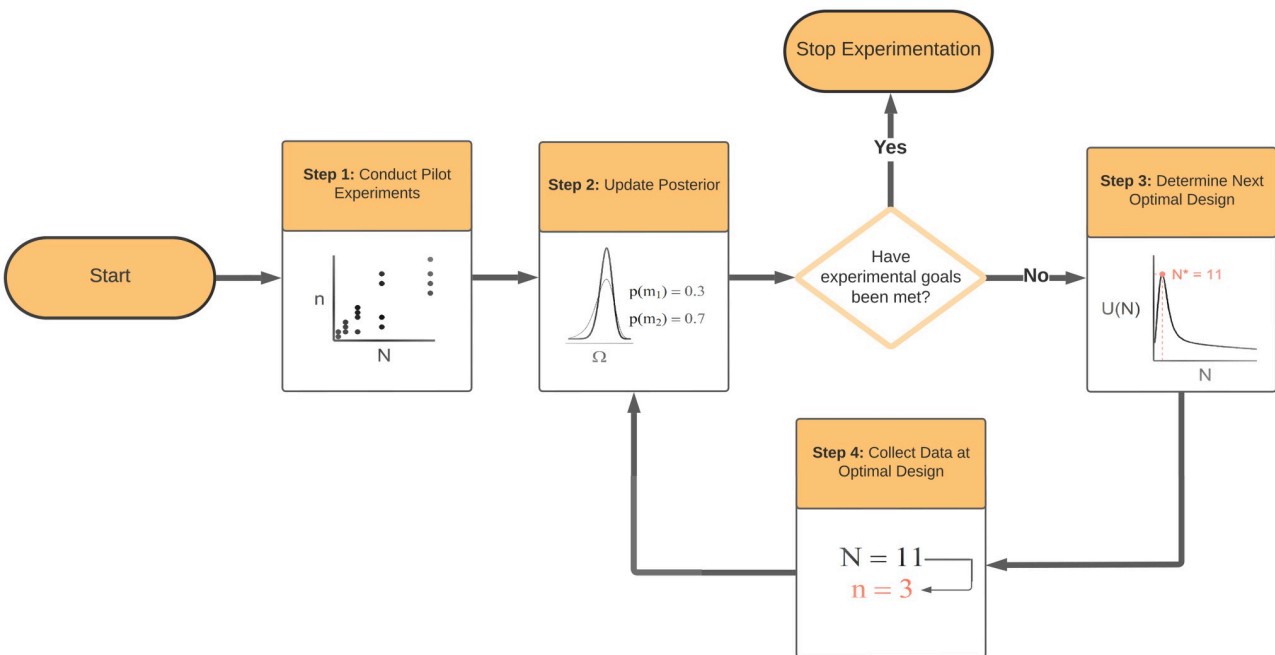

**Fig 2. Workflow for the adaptive experimental design framework.** In Step 1 some pilot experiments are conducted by varying the prey density $N$ and measuring the number of prey eaten $n$. The pilot data is combined with the prior information to obtain the posterior distribution for the model parameters defined on the space $\Omega$ and model probabilities ($p(m_1)$ and $p(m_2)$) in Step 2. Assuming that the experimental goals have not been met with the pilot data, the adaptive design process begins in Step 3 where the optimal prey density ($N^*$) to use for the next experiment is determined by maximising the utility function $U(N)$ with respect to $N$. In Step 4 the experiment is conducted using the prey density found from Step 3. The data collected from Step 4 is then fed back to Step 2 to update the posterior distributions. We again check if the experimental goals have been collected or if the experimental resources have been consumed. If so, we stop the experiment and report the final Bayesian inferences, otherwise we continue to iterative the adaptive design process (Steps 3, 4 and 2 in that order).

For our adaptive experimental design framework (Fig 2), suppose that we have available data up to experiment $t$, $(N_0^{1:t}, n^{1:t})$, and we wish to determine the optimal value of $N_0^{t+1}$ to use for experiment $t + 1$. Our methods are Bayesian, and we thus treat the model indicator as well as the parameters of each of the models as random variables. Before collecting any data, we allocate these quantities prior distributions. After each observation is collected, we produce estimates of the posterior model probabilities for each model and the posterior distributions for the parameters of each candidate model. Based on these current posterior distributions, we find the value $N_0^{t+1}$ by solving the following optimization problem

$$N_0^{t+1} \quad = \quad \arg\max_{N_0 \in \boldsymbol{N_0}} U(N_0 | N_0^{1:t}, n^{1:t}),$$

where $\boldsymbol{N_0} = \{1, 2, \ldots, N_0^{\max}\}$ is the set of allowable prey densities for a particular experiment and $N_0^{\max}$ is the maximum allowable prey density, determined by the experimental system. Here, $U(N_0 | N_0^{1:t}, n^{1:t})$ is called the utility function, which encodes the goal of the experiment. In this paper, and as considered by [26], we use a dual-purpose utility function called the total entropy criterion [29] that aims to simultaneously learn the true model and parameter values as quickly as possible. Basically, we want to learn as much as possible about which model is responsible for data generation and its corresponding parameter values from the next experiment, based on information collected from earlier experiments. Technical details on the utility function are provided in the S1 Appendix. This utility function is the sum of expected

information gains about random variables describing the model indicator and the parameters of each model (see the S1 Appendix for technical details).

The utility function is averaged over all data that we might observe if $N_0$ was chosen for the next experiment, based on the information available prior to the experiment. Then, we search for the value of $N_0$ that maximizes this expected change. That is, we specifically undertake the experiment that—according to our current understanding of the system—will best help us, on average, to differentiate between the available models and simultaneously learn about the parameters of the preferred model. We continue iteratively with independent experiments using the above adaptive experimental design procedure until all $T$ experiments are run, or if some stopping rule is met (see Fig 2). For example, we might stop the experimentation if the maximum posterior probability of one of the functional response models is at least 0.95 and its corresponding parameters are sufficiently well estimated, for example if the 95% credible interval of each parameter is narrow enough. The technical details of our statistical methods for implementing our adaptive design procedure can be found in the S1 Appendix. Bayesian adaptive experimental design methods have been comprehensively studied in the statistical literature (see [30] for a review), and are commonly used in clinical trials (e.g. [31]).

### Assessing the relative performance of adaptive experimental design

As in the baseline experimental design, for our adaptive experimental design method we also conduct 40 experiments for comparison purposes. However, as we show in the results below, we can achieve the similar statistical inferences as the baseline experimental design with much fewer experiments. Matlab computer code for implementing the adaptive design procedure for this study can be found at https://github.com/cdrovandi/Adaptive-Design-Functional-Response. This repository also contains the code and data required to reproduce the results of our paper.

For our optimal adaptive experimental design, we first perform 16 pilot experiments to gain initial information about the two mathematical models under consideration. For notational simplicity we denote Model 1 as the Type II beta-binomial model and Model 2 as the Type III beta-binomial model. This pilot uses 8 experiments at a prey density of 2, and 8 experiments at a prey density of 40. These are a subset of the experiments use in the baseline experimental design, and for simplicity we use the same data collected at these prey levels of 2 and 40 for both designs. We note that different choices for the design of the pilot experiments could be made, in terms of the number of pilot experiments and prey levels. The idea is that the number of pilot experiments should be small, since these choices have been made in a non-optimal way. After the pilot data are collected and the posterior distributions are updated, we run the adaptive experimental design for a further 24 experiments, delivering the same sample size (40 samples) as the baseline experimental design.

To assess how well the data produced by each experimental design method is able to learn about the functional response model parameters, we use the log determinant of the posterior covariance matrix (see, e.g. equation (7) of [32]) recorded after each experiment for each experimental design method as a proxy for how much is learnt about the parameters of each individual model. Here we use a different measurement of parameter information than used in the utility function for validation purposes.

### Results

The data produced by the baseline experimental design are shown in panels (a) and (c) of Fig 3. For the pilot data of the adaptive experimental design, the prey levels $N_0^{1:16} = (2, 2, 2, 2, 2, 2, 2, 2, 40, 40, 40, 40, 40, 40, 40, 40)$ produce response values of $n^{1:16} = (1, 2, 2, 0,$

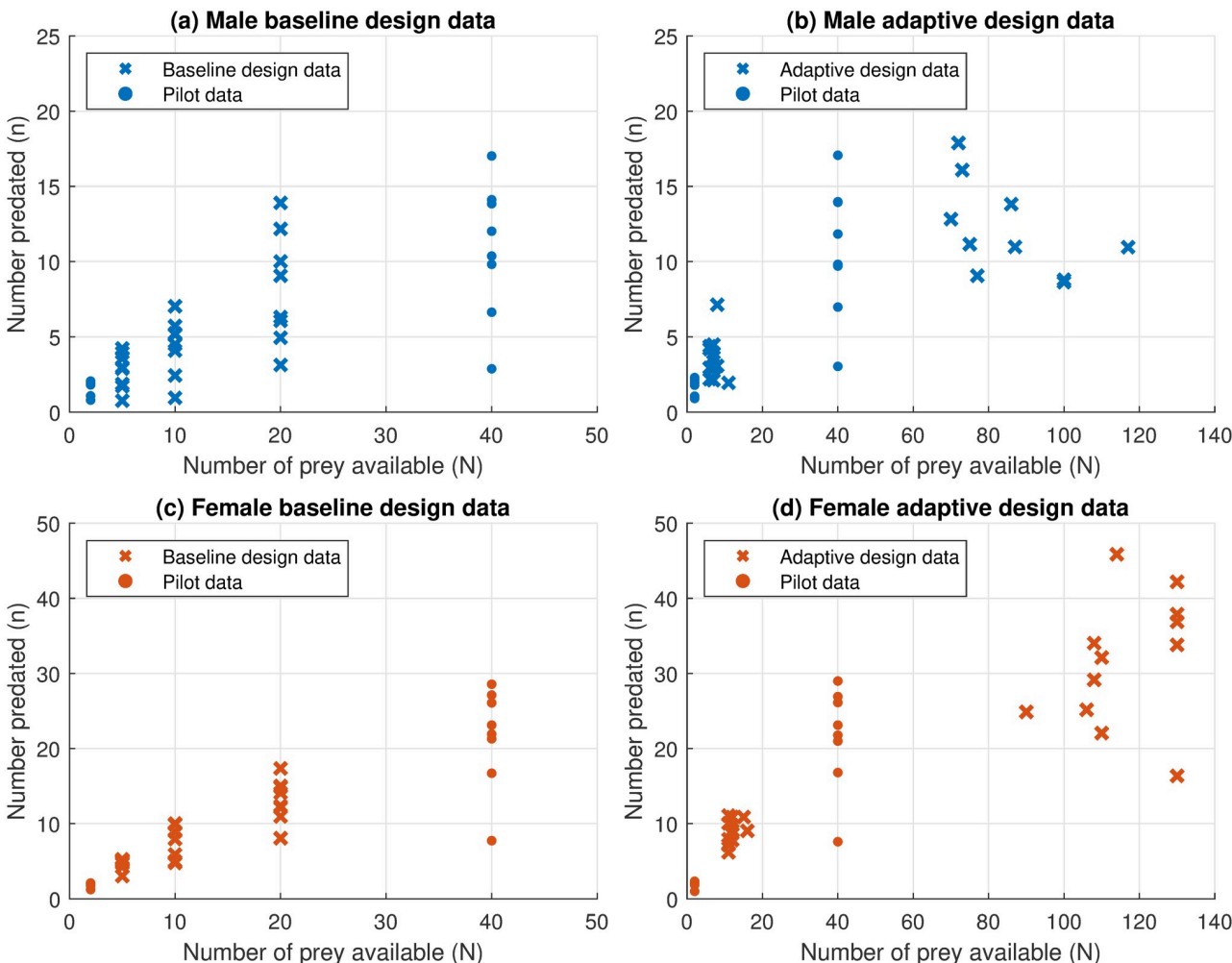

**Fig 3. A visualisation of the raw data collected.** Shown are (a) the data collected for males using the baseline design, (b) the data collected for males using the adaptive design, (c) the data collected for females using the baseline design and (d) the data collected for females using the adaptive design.

2, 1, 2, 2, 14, 7, 10, 3, 12, 14, 17, 10) for males and $n^{1:16}$ = (2, 2, 0, 2, 2, 2, 1, 2, 27, 23, 29, 8, 17, 26, 21, 22) for females. Performing Bayesian inference on the pilot data produces the estimated posterior distributions shown in Fig 4. The posterior model probabilities for males and females are 0.581 and 0.559 respectively, in favour of Model 1 (since there are only two models). Thus, there remains substantial uncertainty in the preferred model and its parameters.

We then undertake the same experiment following our adaptive experimental design process. After each experiment is conducted with a certain prey level (as determined by our adaptive design method), we record the number of prey attacked, and use Bayesian inference to update the posterior distributions of the parameters of the two models and update the posterior model probabilities. The experimental actions recommended by this adaptive design procedure for males and females are shown in panels (b) and (d) of Fig 3, respectively. In contrast to the baseline experimental experiments, the adaptive experiments undertake sampling at different abundances between males and females, and choose prey abundances at very low and very high values of $N$.

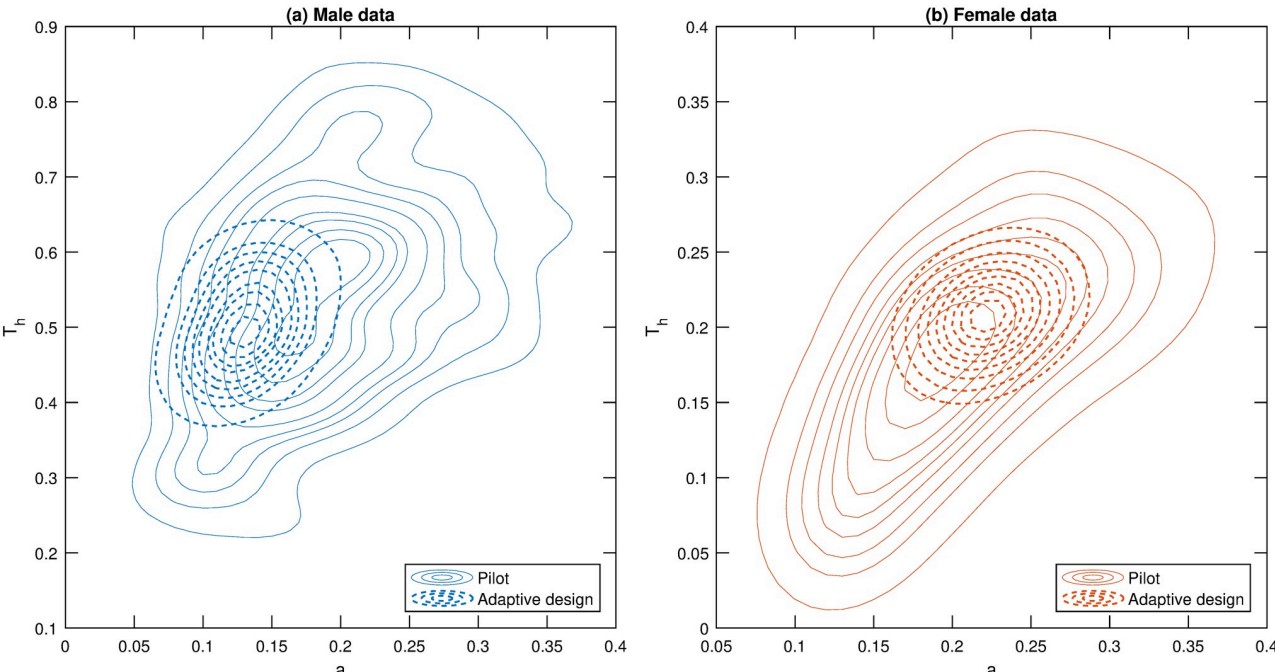

**Fig 4. Estimated posterior distributions from pilot data.** Contour plots (thinned solid lines) estimated from the the bivariate posterior distribution of parameters of Model 1 after the pilot experiments (16 experiments), that is, based on the data collected using the experimental design $N_0^{1:16} = (2, 2, 2, 2, 2, 2, 2, 2, 40, 40, 40, 40, 40, 40, 40, 40)$. The area enclosed within the outer contour represents an estimated 90% credible region (i.e. a 90% probability that the region contains the true parameter value), and the area enclosed in each contour within drops the credible region % by 10% so that the area enclosed within the inner contour represents a 10% credible region. Panels (a) and (b) are results based on data collected for males and females, respectively. As a reference, the contour plots estimated from the bivariate posterior distribution applying our adaptive design methodology is shown as thick dashed lines.

Fig 5 illustrates how rapidly we learn about the preferred model (panel (a) of Fig 5) and the model parameters (panel (b) of Fig 5). It is evident that the adaptive design learns that Model 1 is clearly the preferred model with substantially fewer experiments, compared to the baseline experimental design. Following the collection of the pilot data, the adaptive experimental design process only requires an additional 3 observations to achieve a posterior model probability greater than 0.95 for both male and female data. In comparison, the baseline experimental design needs another 17 data points to arrive at similar posterior model probabilities. For this preferred model, it is also clear that adaptive design provides more information about the model parameters, more rapidly, compared with baseline experimental design. For males, the adaptive experimental design achieves the same parameter precision as the full set of baseline experimental data with only 12 additional samples. For females, only 5 additional samples are required to reach the same level of precision as the full set of baseline experimental data. Fig 6 compares the bivariate posterior distributions of the parameters after all the data have been processed for the adaptive and baseline experimental designs. The results show that the adaptive design produces much more precise posterior distributions for the parameters.

In terms of comparing whether there is a difference in parameter values between males and females, it can be seen from Fig 6 that there is little overlap in the 90% credible regions between males and females, regardless of the experimental design used (i.e. baseline or adaptive). Thus, for this predator-prey system, there is strong evidence of a difference in parameter values between males and females. It is important to note that in this study we did not explicitly aim to design for testing if there was a difference in parameter values between males and females;

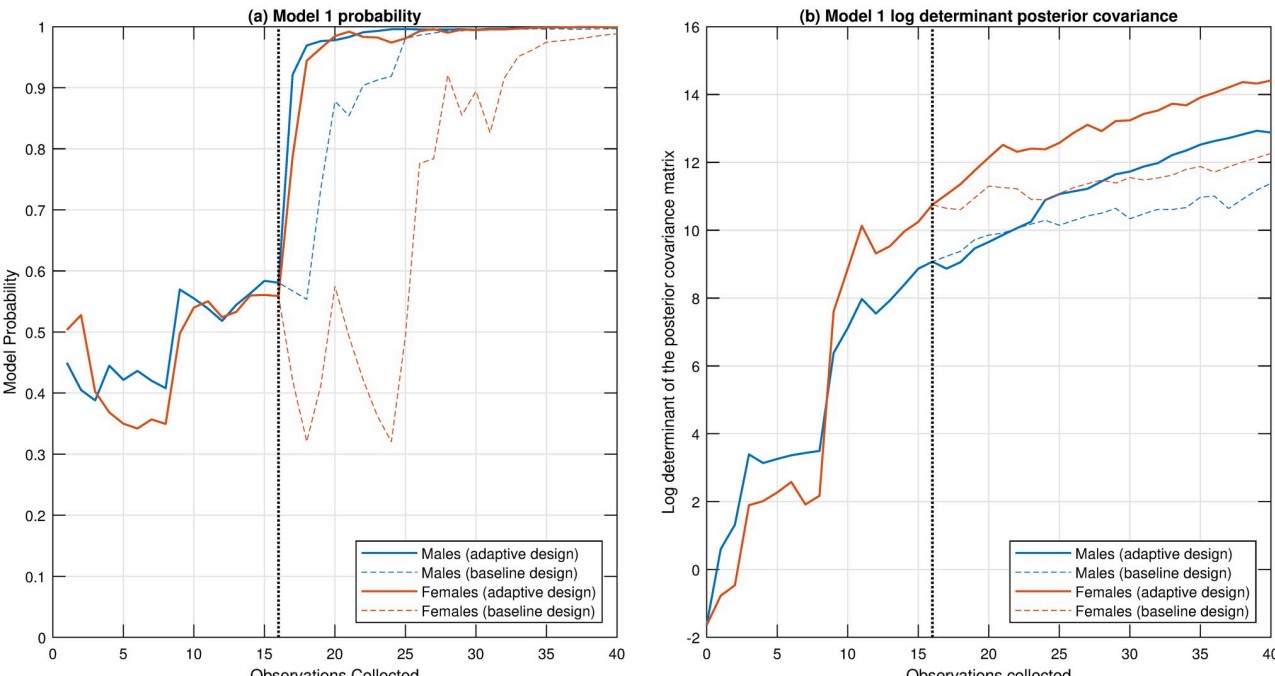

**Fig 5. A comparison of performance between the two experimental design approaches.** Shown are results for (a) models selection and (b) parameter estimation for the preferred model. The black dotted line in each of the subfigures indicates the conclusion of the pilot experiments.

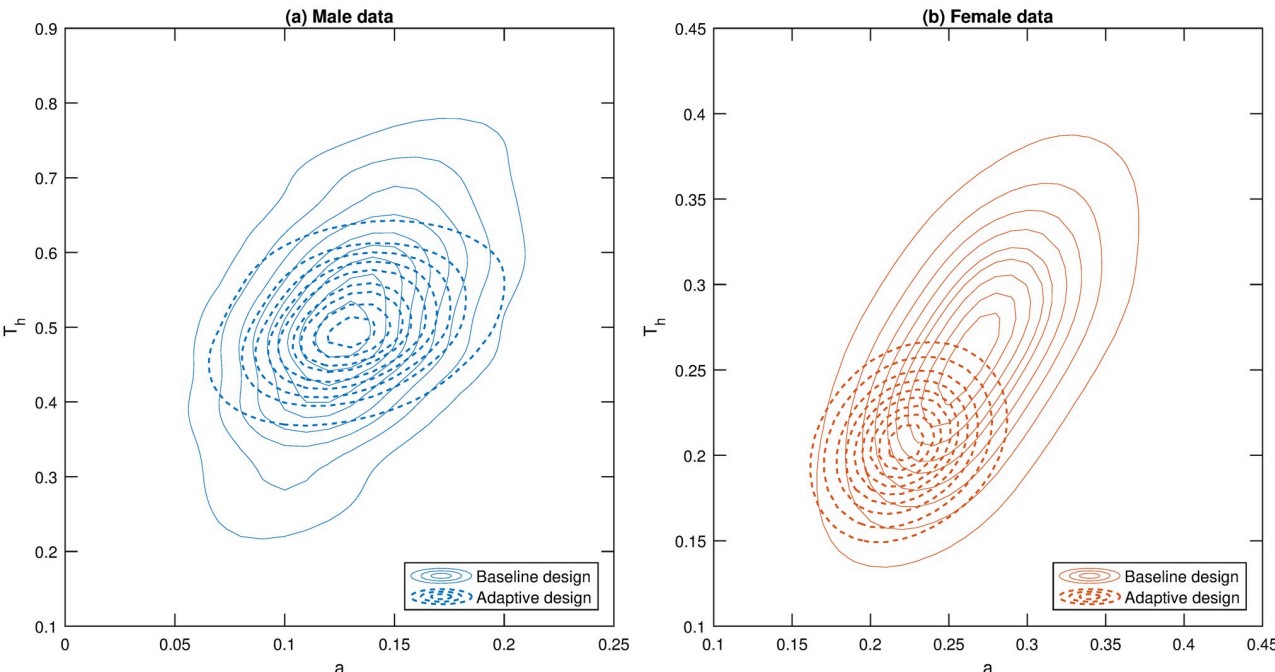

**Fig 6. Estimated posterior distributions for baseline and adaptive designs.** Contour plots estimated from the the bivariate posterior distribution of the attack rate (a) and handling time ($T_h$) for the static experimental and adaptive designs. See the caption of Fig 4 for information on the contour lines. The thicker dashed lines represent bivariate posteriors obtained from data using the adaptive design. Panels (a) and (b) represent results for the male and female data, respectively.

we essentially treated males and females as separate studies. If there was interest in designing to distinguish between the parameters of different groups, i.e. males and females, then additional models should be incorporated into the candidate list that assume shared parameter values between groups (as well as the models that allow parameter values to be different between groups).

## Discussion

Distinguishing among competing models is a challenging issue in functional response studies [22, 33, 34]. For example, discriminating between type II and type III functional responses is crucial for the comprehension and prediction of the stability and the dynamics of predator-prey systems [35]. In addition, robust parameter estimation of the selected model that determine predator feeding behavior in relation to prey density—such as the predator attack rate and handling time, is fundamental in order to elucidate and quantify predator-prey interactions (e.g. [34, 36, 37]), and to predict the consequences of conservation actions and species invasions [14, 17]. Our study revealed that the accuracy and consistency of functional response experiments can be improved adopting an adaptive experimental design framework. Thus, taking advantage of our methodology should increase the probability that predator feeding behavior can be described by an appropriate functional response model, followed by precise parameter estimation. In addition, in our case study reduced experimentation was needed, minimizing the number of animals required, compared to a traditionally designed functional response experiment. This could lead to reduced logistical burdens and animal ethics considerations, both factors that can constrain experimentation.

While the specific aim of our analysis was to test whether the functional response is of type II or type III, our adaptive experimental design approach could be potentially used in future research to test several scientific hypotheses concerning predator feeding behavior; that is, to explore if the nature of predation is predator, prey or ratio dependent, which is a core question regarding predator-prey interactions [38–41] or to select the most appropriate functional response model that incorporates mutual interference effects and best describe the data (e.g. [19, 42, 43]). Thus, it is expected that following our experimental framework, the description, understanding and quantitative predictions of predator-prey interactions can be improved, towards a precise understanding and integrated analysis of ecological processes.

This study presented the first demonstration of the benefits of an optimal adaptive experimental design approach to a real functional response experiment. In this study we used 16 observations of pilot experimentation before initiating the adaptive experimental design procedure. In retrospect, this relatively large number of pilot experiments may not be required. With fewer pilot experiments, the sequential process will likely enhance the superiority of adaptive methods over baseline or static experimental decisions, drawing faster and cheaper conclusions about the predator-prey system. In this paper we ran a given number of experiments, but it is possible to terminate the experimentation early if the experimental goals have been met to the satisfaction of the practitioner.

Since each of our microcosm experiments lasted 8 hours, we were able to update our sequential decisions before each new experiment. However, other functional response experiments may run much longer, and it may not be time-feasible to run one experiment at a time. For such experiments it may be of interest to obtain an optimal static experimental design, or an optimal batch experimental design, as opposed to a strictly adaptive design. In a static experimental design, after the initial pilot data are collected, the prey densities are determined for all of the remaining experiments. In a batch design, a subset of the remaining experiments

are planned at the same time. Both alternatives would outperform the regular *a priori* or baseline design we considered here. Neither will be as informative as a strictly adaptive design [26].

This study considered relatively simple stochastic models of predator-prey systems. However, in principle, more complex models can be accommodated in our framework, such as those modelling predator interference [42]. For experiments conducted in different spatial locations, a fixed or random spatial effect can be added to the model. Our experiments were conducted over a short time period. However, for systems with longer experimental times, various conditions and the manner in which the predator and prey interact may change. In which case, it might be necessary to allow parameters to be time-varying, permitting small changes in either continuous or discrete time. We do acknowledge that the optimal approach to experimental design can become more computationally intensive for more complicated models (see [30] for a detailed discussion of the issues). Importantly, the adaptive design approach requires fewer runs to achieve the statistical goals, which can help to mitigate potential temporal effects.

In a narrow sense, our results demonstrate the value of optimal adaptive experimental design for estimating the functional response of predators to prey density. However, the benefits of adaptive experimental design could be pursued across a much broader set of research areas. A static approach to experimental design is not just the standard approach to estimating functional responses—across the field of ecology, experiments are drawn up in advance, and experimental designs are rigidly followed even as incoming data reveals new information about the system. This inflexible approach places ecology behind the modern experimental frontier: in fields as diverse as clinical medicine [44], conservation management [45], and molecular biology [46], scientists are adapting their experimental plans as soon as the data begins to flow. The results of our analyses and experiments demonstrate the very substantial gains that can be achieved by an approach to experimental design that learns as it goes. We hope that the magnitude of these efficiencies encourages more experimental ecologists towards adaptive design, and motivates statistical ecologists to improve on our methods.

## Supporting information

**S1 Appendix. Supplementary document.** pdf of supplementary document referred to in the main text.
(PDF)

## Acknowledgments

Helpful comments from two anonymous referees and the Editor led to improvements in this paper.

This article is dedicated to Dr Nikos E. Papanikolaou, who sadly and unexpectedly passed away after the majority of this manuscript had been completed.

## Author Contributions

**Conceptualization:** Nikos E. Papanikolaou, Christopher Drovandi.

**Formal analysis:** Hayden Moffat, Christopher Drovandi.

**Investigation:** Nikos E. Papanikolaou, Hayden Moffat, Argyro Fantinou, Dionysios P. Perdikis.

**Methodology:** Hayden Moffat, Christopher Drovandi.

**Software:** Hayden Moffat, Christopher Drovandi.

**Supervision:** Michael Bode, Christopher Drovandi.

**Writing – original draft:** Nikos E. Papanikolaou, Hayden Moffat, Michael Bode, Christopher Drovandi.

**Writing – review & editing:** Nikos E. Papanikolaou, Hayden Moffat, Argyro Fantinou, Dionysios P. Perdikis, Michael Bode, Christopher Drovandi.

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
