## [Decision Letter · Decision Letter 0]

20 Mar 2023

PONE-D-23-02754Adaptive experimental design produces superior and more efficient estimates of predator functional responsePLOS ONE

Dear Dr. Drovandi,

Thank you for submitting your manuscript to PLOS ONE. After careful consideration, we feel that it has merit but does not fully meet PLOS ONE’s publication criteria as it currently stands. Therefore, we invite you to submit a revised version of the manuscript that addresses the points raised during the review process.

 As you will see, both Reviewers are positive about your contribution but they also raise a number of comments that could be handled without difficulty. The main issue is raised by Reviewer 2 and concerns the logic of sequential designs. This is a valid remark that, to my opinion, questions more the general approach of sequential designs and not specifically your contribution. However, it is important and useful to explain how Bayesian inference differs from the frequentist approach in dealing with parameter estimation, and potentially how it can account for random factors.

Personally, I read your manuscript with pleasure. I think that your proposed approach has a potential to become a standard since it is more cost-effective than standard designs. It is useful and general as it could accommodate more complex functional responses, for example with predator interference.   

We look forward to receiving your revised manuscript.

Kind regards,

Louis-Felix Bersier, Ph.D.

Academic Editor

PLOS ONE

Journal Requirements:

Reviewers' comments:

Reviewer's Responses to Questions

**Comments to the Author**

1. Is the manuscript technically sound, and do the data support the conclusions?

Reviewer #1: Yes

Reviewer #2: Partly

2. Has the statistical analysis been performed appropriately and rigorously? 

Reviewer #1: I Don't Know

Reviewer #2: Yes

3. Have the authors made all data underlying the findings in their manuscript fully available?

Reviewer #1: Yes

Reviewer #2: No

4. Is the manuscript presented in an intelligible fashion and written in standard English?

Reviewer #1: Yes

Reviewer #2: Yes

5. Review Comments to the Author

Reviewer #1: PONE-D-23-02754

Adaptive experimental design produces superior and more efficient estimates of predator functional response

This paper describes a procedure for optimal design of functional response experiments, designed to achieve best estimates with the fewest number of experiments / replicates, using Bayesian inference and Monte Carlo simulations. The procedures are derived from previous papers (particularly: Moffat H, et al. 2020 Sequential experimental design for predator–prey functional response experiments. J. R. Soc. Interface 17: 20200156. http://dx.doi.org/10.1098/rsif.2020.0156, and related empirical papers). The “adaptive” aspect of the design involves using preliminary experiments sequentially to improve model selection or estimation by not only adding more replicates but optimally choosing the levels of the independent variable (number of prey offered) that most efficiently achieve the goals of the study. They paper advances what is shown in that previous paper by illustrating how the procedure works with real experimental data, as opposed to simulation that were used in the previous paper. The current paper shows how the goals of experiments may be met with fewer experiments, and how estimates may be improved using this adaptive experimental designs.

My background is that of an ecologist with some background in applied, non-Bayesian statistics, but limited background in mathematical / statistical theory. Hence, I will have relatively little to say about the technical aspects of the optimization methods used and the mathematical basis of this approach. However, the overall goal of more efficient experiments is an important one for empiricists, and I believe this paper makes a sound case for the utility of the adaptive approach. Some of my suggestions are directed at making the presentation clearer, particularly for those who may not have the strong mathematical background evident in the paper and supplement, but who nevertheless are interested in using experiments on functional responses to answer biological questions. My hope would be that those suggestion would make it more likely that this adaptive approach becomes commonly used.

My comments are divided in to major and minor ones, with the latter being mostly about wording and typos.

Major comments.

Line 128-129. Regarding p(tau). This statement would mean that p(tau) = proportion of prey eaten for experiment of duration tau, p(tau) =n/N0. Correct?

Lines 138-141. When I first read this, I interpreted it to mean that the independence assumption means that the experiments are conducted individually, in sequence, at different times, using a single N0 at each time. Based on later statements (Line 190), I believe I read too much into this description. Sequential, at different times, but flexible about the number of different N0 values. Think you could make your meaning clearer here.

Line 213, Figure 4. This figure needs a better, more explicit caption. Is there any connection between these posterior distributions and the bivariate distributions shown in Fig. 6? See further comment on Figure 6.

Lines 222-223. In addition to deciding on the model most appropriate for a predator's feeding (type II or III), and estimating that model's parameters, another goal of many functional response studies is to compare statistically 2 or more functional responses to test the hypotheses that one or more FR parameters differ among the groups (e.g., males vs. females). My guess is that if your approach produces better estimates, then it should produce better, more statistically powerful, comparisons. I think you should address this issue if you can. It seems like this comparison of males vs. females in the data you use would allow you to address that issue.

If you prefer to cast this in a model selection context, rather than hypothesis testing, then address whether your approach improves your ability to decide e.g., whether a full model with same parameters (a, Th) for males and females is preferable to models postulating that one or both parameters (a, Th) have different values for males and females, as indicated by model likelihood or information criteria. My main point is that COMPARISONS are another issue your paper should address.

Line 239, Figure 6. I think Figure 6 needs greater explanation, particularly exactly what the concentric isoclines represent - Empirical distributions? Bootstrapped values? confidence intervals? A more detailed explanation will make your point more clearly even to people who don't think Bayesian

Figure 6 caption. For males this appears to be centered on Th=0.2 and a=0.22 That does not seem to match well with what is shown in Figure 4 for the univariate distributions after the pilot expt. Is that because these estimates improved (and shifted) compared to the pilot? The same mismatch is evident in the female data. I think your point would be clearer if you showed Figure 4 (after pilot) and Figure 6 (Final) distributions the same way, and likely side by side. I like the bivariate form. This would make it clearer what the reader is seeing and showing the improvement of the estimates that occurs via the adaptive design. You might show that on the same graph, but that would need to be separated from the comparison of adaptive vs. static.

Discussion and Supplementary material. I think that in some cases, running the experiments sequentially may create a confounding problem, if the predators, prey, conditions, or experimenters vary over time. That would mean that different prey densities (implemented in experiments at different times) would be confounded with different time effects. Whether this is a real problem, and how serious it is would likely depend on the details of the experiment (the organisms, the set up, and how far apart in time the experiments are run). Have you considered this issue and is there a way to detect it while implementing the optimal design? The different iterations of the experiments performed at different times seem to me to be equivalent to temporal blocks. Could the optimal design incorporate something like temporal blocks if the time effect does prove to be a non-trivial issue?

I suspect that for the system you describe here, this is not much of an issue, with duration of only 8 h, and fairly short-lived and likely numerous prey and predators. But in other systems, with longer experimental times, or prey / predators that are more difficult to acquire in large numbers (which might actually be the kinds of organisms where this more efficient design is most valuable) the temporal effect might be more important. Some discussion of this issue would be useful, in the main paper.

Supplementary material. My mathematical expertise does not allow me to critique the programming or modeling approach used here. The code mentioned in the preliminary information in the manuscript itself will be very useful in enabling ecologists with lesser mathematical / coding experience to implement these methods. The Github code sets (for R and Matlab) from the previous paper by Moffat et al. are a good model for the kind of detail that would help experimentalists. The more you can provide, the more often and widely your approach will be implemented.

Minor comments.

Line 68. Spelling. The name is “Stouffer”

Line 80. 'among' would be a better word. A restructured sentence might be clearer: "...that it would be challenging to discriminate among these three functional forms..."

Line 119. Th . check your symbols.

Line 198 (and multiple other places, e.g., line 285). “data” is a plural, so ‘are’ should replace ‘is’.

Line 209. Missing figure number. Figure 4 is what you mean here, I believe. This also comes up elsewhere (line 228, 239), perhaps as a problem translating to PDF?

Line 240. ‘have’ not ‘has’

Reviewer #2: This is a very well written and clear manuscript. The authors do an excellent job of explaining and complex and potentially powerful approach. I find the idea of adaptive experimental designs intriguing and potentially powerful. I was particularly compelled by the idea that this could reduce the number of animals needed for experiments. However, there are a few places where I think the manuscript could be made clearer and a few big statistical questions that still need to be addressed.

Line 74-76: I do not think this description of a typical experimental design is accurate. I would argue that most studies use a static experimental design where a set of predators are provided one of several densities and all density treatments are conducted simultaneously. The iterative approach described is not the typical method. More justification of this is needed or clarification of when such designs are used relative to more typical experimental designs.

Line 79-81: I feel that this example is a bit of a straw man. We do not typically use functional responses to distinguish models at a single high or low density as depicted in Figure 1. Instead, inferences are more typically based on the entire Functional response curve, or the curves are used to make projections. In the latter case predictions at density B would be indistinguishable and the implications minimal for population dynamics.

Material and Methods:

The description of the functional response experiments on line 102 – 110 was inadequate. It was not clear that there were eggs on the tomato leaflet, what densities were used, and how many replicates. This information is not presented until > 70 lines later. I think the design and contrasts would be more clear if the presentation was reorganized to highlight the baseline and pilot design first and then describe the adaptive methodology. Or describe the method first and then the full design used to test it.

Discussion:

I feel like this paper is incomplete without formal analysis of the approaches suggested on lines 284 – 288. These intermediate designs are most relevant for most researchers and systems. This might also allay some of the concerns outlined in the next section.

General Comment:

My biggest concern with the presentation is the lack of statistical exploration and explanation. In a typical static experimental design, all density treatments are carried out simultaneously and are presumable randomly assigned to predators. The entire set of density treatment may then be replicated in multiple replicate blocks. The effects of time, spatial location, or other variable that might have affected the specific set of predators and prey used in a replicate can be estimated and accounted for as a random effect. How does this piecemeal approach estimate and propagate these types of errors, and the nonrandom assignment of treatments?

These issues became particularly prescient for me as I examined Figure 2. This sequential adaptive process looks alarmingly like a phenomenon commonly referred to as p-hacking in traditional hypothesis testing. How is this method different from ad hoc experimental design which occurs when you did not choose a sample size a priori, but instead keep doing replications until you get desired results. I realize the proposed methods are much more sophisticated than a simple hypothesis test. However, it still seems as though there needs to be some sort of penalty that you must pay for the adaptive “adaptive“ sample size decisions prior to declaring a “significant” or best model.

6. PLOS authors have the option to publish the peer review history of their article (what does this mean?). If published, this will include your full peer review and any attached files.

Reviewer #1: No

Reviewer #2: No

---

## [Author Response · Author response to Decision Letter 0]

19 May 2023

Point-by-point response to referees and editor has been attached as a separate pdf file.

---

## [Decision Letter · Decision Letter 1]

21 Jun 2023

PONE-D-23-02754R1Adaptive experimental design produces superior and more efficient estimates of predator functional responsePLOS ONE

Dear Dr. Drovandi,

Thank you for submitting your manuscript to PLOS ONE. After careful consideration, we feel that it has merit but does not fully meet PLOS ONE’s publication criteria as it currently stands. Therefore, we invite you to submit a revised version of the manuscript that addresses the points raised during the review process. As you will see, the Reviewer #2 is happy with your revision and suggested its acceptance. The external Reviewer #1 could not re-examine your contribution and I decided to do it. I found several small problems, and they need to be corrected for your text to be publishable. You will find my comments below. Once these corrections are done, I will be able to reach a decision without further help of the Reviewers. Overall, I think that your contribution will be very useful for experimental ecologists. I read it with pleasure. 

We look forward to receiving your revised manuscript.

Kind regards,

Louis-Felix Bersier, Ph.D.

Academic Editor

PLOS ONE

Journal Requirements:

Additional Editor Comments:

I have a list of minor comments to your Revision 1:

1) Legend of Fig. 2: I would be clearer with the abbreviations of the figure (a figure should be understandable without reference to the main text). I suggest:

"In Step 1 some pilot experiments are conducted by varying the prey density N and measuring the number of prey eaten n. They are combined with the prior information to obtain the posterior distribution for the model parameters (Omega = m1 and m2) and model probabilities (p) in Step 2. Assuming that the experimental goals have not been met with the pilot data, the adaptive design process begins in Step 3 where the optimal prey density (N) to use for the next experiment is determined by maximising the utility function U(N). In Step 4 the experiment is conducted using the prey density found from Step 3. The data collected from Step 4 is then fed back to Step 2 to update the posterior distributions. We again check if the experimental goals have been collected or if the experimental resources have been consumed. If so, we stop the experiment and report the final Bayesian inferences, otherwise we continue to iterative the adaptive design process (Steps 3, 4 and 2 in that order)."

Please check my proposition for consistency and format if necessary. Also, in the box for Step 4, I would add a vertical arrow from N=11 pointing to n=3 to make evident that n=3 is the result of the experiment.

2) line 183: Remove the second of in "...the number of pilot of experiments."

3) lines 188 to 193: I would provide a reference for the "log determinant of the posterior covariance matrix", as this concept may be unknown for the majority of readers (it was for me).

4) line 195: Remove "produces data" in "The data produced by the baseline experimental design produces data are shown in Figs..."

5) lines 200-202: "The posterior model probabilities for males and females are 0.581 and 0.559 respectively, in favour of a Type II functional response."

First, I would use "model 1" and "model 2" everywhere in the Result section. Second, it is not clear to me why the model 1 is favoured (see also lines 214-215). It seems to me that the information should be given in Fig. 5, but I do not see the results for model 2 in this figure. Please clarify this issue.

6) line 209 and Fig. 3: please check the identity of the four panels in the text and in the figure. Moreover, the panel titles do not correspond to the legends.

7) Legend of Fig. 4 and Fig. 6: The "top and bottom rows" do not correspond to the figures. Please use instead "panel a" and "panel b".

8) Legend of Fig. 5: Please correct the first sentence ("Shows are results for...").

9) Legend of Fig. 6: Rather than repeating the long text of the legend of Fig. 4, I would simply use "See the legend of Fig. 4 for information on the contour lines".

Reviewers' comments:

Reviewer's Responses to Questions

**Comments to the Author**

1. If the authors have adequately addressed your comments raised in a previous round of review and you feel that this manuscript is now acceptable for publication, you may indicate that here to bypass the “Comments to the Author” section, enter your conflict of interest statement in the “Confidential to Editor” section, and submit your "Accept" recommendation.

Reviewer #2: All comments have been addressed

2. Is the manuscript technically sound, and do the data support the conclusions?

Reviewer #2: Yes

3. Has the statistical analysis been performed appropriately and rigorously? 

Reviewer #2: Yes

4. Have the authors made all data underlying the findings in their manuscript fully available?

Reviewer #2: Yes

5. Is the manuscript presented in an intelligible fashion and written in standard English?

Reviewer #2: Yes

6. Review Comments to the Author

Reviewer #2: Thank you for your thoughtful consideration of the comments and your responses. I think this is a very interesting contribution to the field.

7. PLOS authors have the option to publish the peer review history of their article (what does this mean?). If published, this will include your full peer review and any attached files.

Reviewer #2: No

---

## [Editor Report · Decision Letter 2]

28 Jun 2023

Adaptive experimental design produces superior and more efficient estimates of predator functional response

PONE-D-23-02754R2

Dear Dr. Drovandi,

We’re pleased to inform you that your manuscript has been judged scientifically suitable for publication and will be formally accepted for publication once it meets all outstanding technical requirements.

Kind regards,

Louis-Felix Bersier, Ph.D.

Academic Editor

PLOS ONE

Additional Editor Comments (optional):

Please check again the titles of the 4 panels of Fig.3 (According to the data points and the legend, I think that panel a should be "Male adaptive design data", panel b should be "Male baseline design data", and similarly for the panels c and d). Correct the figure if necessary, and check the main text to be sure that the references to these panels are coherent.
---

## [Editor Report · Acceptance letter]

12 Jul 2023

PONE-D-23-02754R2 

Adaptive experimental design produces superior and more efficient estimates of predator functional response 

Dear Dr. Drovandi:

I'm pleased to inform you that your manuscript has been deemed suitable for publication in PLOS ONE. Congratulations! Your manuscript is now with our production department. 

Kind regards, 

on behalf of

Prof Louis-Felix Bersier 

Academic Editor

PLOS ONE